# Compositional Condition Question Answering in Tabular Understanding

**Jun-Peng Jiang** [1 2]   **Tao Zhou** [1 2]   **De-Chuan Zhan** [1 2]   **Han-Jia Ye** [1 2]

## Abstract

Multimodal Large Language Models (MLLMs) for tabular understanding have made significant progress in tasks such as financial report analysis and public data tests. However, our comprehensive analysis shows that these models are still limited in certain simple scenarios, particularly when handling compositional conditions in QA. Further investigation reveals that the poor performance can be attributed to two main challenges: the visual encoder's inability to accurately recognize the content of a row, and the model's tendency to overlook conditions in the question. To address these, we introduce a new Compositional Condition Tabular Understanding method, called CoCoTab. Specifically, to capture the structural relationships within tables, we enhance the visual encoder with additional row and column patches. Moreover, we introduce the conditional tokens between the visual patches and query embeddings, ensuring the model focuses on relevant parts of the table according to the conditions specified in the query. Additionally, we also introduce the Massive Multimodal Tabular Understanding (MMTU) benchmark, which comprehensively assesses the full capabilities of MLLMs in tabular understanding. Our proposed method achieves state-of-the-art performance on both existing tabular understanding benchmarks and MMTU. Our code can be available at https://github.com/LAMDA-Tabular/MMTU.

## 1. Introduction

Tabular understanding (Deng et al., 2022; Jin et al., 2022; Shigarov, 2023; Wan et al., 2024) aims to automatically extract, analyze, and comprehend information from various types of tables, such as images or screenshots. It is a critical task in numerous fields, including financial analysis (Zavitsanos et al., 2024), experimental data interpretation (Konopka et al., 2023), and public service record management (Engin & Treleaven, 2019).

Recent advancements in Multimodal Large Language Models (MLLMs) (Zhu et al., 2023; Achiam et al., 2023; Liu et al., 2024b;a; Yin et al., 2023) have significantly progressed the field of tabular understanding. These models, which integrate pre-trained large language models (LLMs) with vision encoders through projections, provide an end-to-end framework. While conventional tabular understanding tasks, including table detection and structure recognition, have been successfully addressed by previous studies (Xu et al., 2020; Hu et al., 2021a; Huang et al., 2022; Tang et al., 2023; Wei et al., 2024), our work focuses on the more challenging task of table question answering (TQA), particularly in scenarios where tabular data is presented in image format.

For simple TQA tasks, MLLMs may only need to recognize and interpret the table's structure to provide accurate answers. For instance, in a grade report, MLLMs can easily extract a specific student's score for a course. However, such tasks do not engage with the table's inherent structure, where rows represent samples and columns represent attributes. The complexity of rows and columns often leads to compositional queries (Oh et al., 2011; Lei et al., 2018; Talmor et al., 2021; Dang et al., 2024), like identifying the mathematics score of the student with the highest overall total. Such tasks demand a more profound understanding of the table's content and structure, significantly increasing the complexity and requiring enhanced interpretative capabilities from MLLMs. This raises the question: When provided with a table image, can MLLMs understand the structural information and handle such complex TQA tasks?

To answer the question, we conducted a preliminary investigation. Based on the characteristics of tabular data, we categorize the existing benchmark into four aspects: understanding individual elements (IE), interpreting rows and columns (RC), comprehending compositional conditions (CC), and performing basic calculations or reasoning (CR). Our analysis, shown in Figure 1, reveals that current MLLMs perform well on IE and RC tasks, which mainly require structural recognition. However, they face significant limitations on

[1]School of Artificial Intelligence, Nanjing University, Nanjing, China. [2]National Key Laboratory for Novel Software Technology, Nanjing University, Nanjing, China. Correspondence to: Han-Jia Ye <yehj@lamda.nju.edu.cn>.

*Proceedings of the 42nd International Conference on Machine Learning*, Vancouver, Canada. PMLR 267, 2025. Copyright 2025 by the author(s).

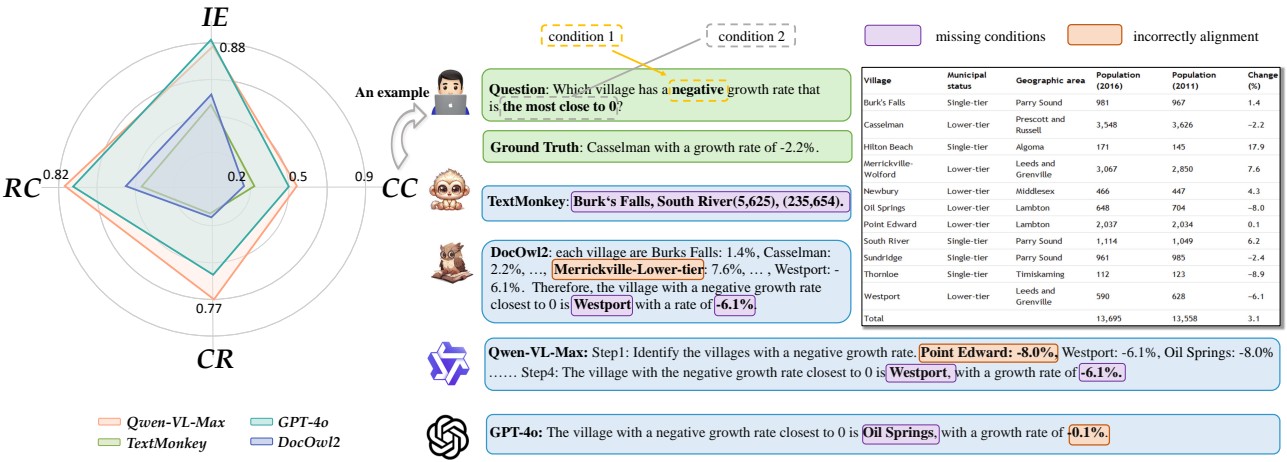

*Figure 1.* An overview of the capabilities of some current MLLMs on a subsample of WikiTableQuestions (Pasupat & Liang, 2015) dataset in the four aspects of tabular understanding, including understanding individual elements (IE), rows or columns (RC), compositional conditions(CC), and calculations or reasoning (CR). An example on the CC is also provided. Almost all models perform poorly, with the two main types of errors being: incorrectly aligned with the row or column, and overlooking a condition specified in the question.

CC and CR tasks, which require deeper engagement with the information in rows and columns. MLLM performance varies notably, even on simple IE tasks. Furthermore, all models, including GPT-4o (OpenAI, 2024), perform poorly on tasks involving basic compositional conditions.

To understand what exactly causes current MLLMs to perform poorly in tabular understanding, especially in compositional conditions. we investigate several aspects according to the three components of MLLMs: vision encoder, projection, and LLM. Our analyses suggest that the primary reason for the observed poor performance is the vision encoder and the projection. Specifically, the patch division strategy of the vision encoder causes an inability to accurately recognize the content of a row, leading to positional errors in the table's structure. Moreover, when handling complex QA, MLLMs tend to overlook all the relevant conditions in the question, leading to incomplete or incorrect answers.

To address this, we propose Compositional Condition Tabular Understanding (CoCoTAB), a new MLLM aimed at improving tabular understanding. In CoCoTAB, the visual encoder is enhanced to better capture the full structure of tables, processing row and column information without the distortions from dividing the table image into patches. We also improve the alignment between the table's structure and the query conditions, ensuring key information is attended. Our approach offers a lightweight enhancement to the original model without adding complexity.

Specifically, CoCoTAB works by augmenting the visual encoder with row and column patches, which allows the model to better capture the structural relationships within tables. These patches provide contextual information in specific rows or columns, helping the model focus on the specific sample or attribute of the table, and improving its

understanding of contextual dependencies. Moreover, we introduce the conditional tokens between the visual patches and query embeddings. This ensures that the model can focus on the relevant parts of the table based on the conditions specified in the question, allowing it to effectively reason across different rows, columns, or conditions. Experimental results demonstrate the effectiveness of CoCoTAB in improving the model's ability in complex TQA tasks, particularly those involving compositional conditions.

To better assess MLLMs' ability in tabular understanding, we identify gaps in current benchmarks, such as a lack of focus on specific question types and the presence of factual errors. We propose evaluating four key aspects: element, row/column, compositional condition understanding, and basic calculations/reasoning. Introducing the Multimodal Tabular Understanding (MMTU) benchmark, with 8921 QA pairs across 4 categories and over 10 domains, we aim to overcome existing dataset limitations. The key contributions of this paper are as follows:

- **Evaluating MLLMs weaknesses in tabular understanding:** Our evaluation with multiple MLLMs on benchmarks demonstrates the significant disparities and limitations of MLLMs in tabular understanding.
- **Analyzing causes of limited understanding performance:** Through testing various hypotheses, we find that the primary reasons for MLLMs' poor performance in tabular understanding are misalignment in vision encoders and overlook of conditions in questions.
- **Enhancing MLLMs with structural information:** By incorporating row and column patches and attention mechanisms, we improved the performance of MLLMs in tabular understanding. The new benchmark also provides a more comprehensive evaluation.

## 2. Related Work

### 2.1. Mutlimodal Large Language Models

The field of multimodal large language models (MLLMs) has seen remarkable progress, especially in the integration of visual and textual processing. Modern MLLMs typically combine visual encoders (Radford et al., 2021; Sun et al., 2023; Zhai et al., 2023; Han et al., 2022), large language models (LLMs) (Brown et al., 2020; Ouyang et al., 2022; Chang et al., 2024; Tianhao et al., 2025; Lian et al., 2025), and various fusion modules (Li et al., 2024; Sun et al., 2025b;a; Zhang et al., 2024; 2025). Recent developments, such as Flamingo (Alayrac et al., 2022), have advanced visual representation by utilizing the Perceiver Resampler alongside vision encoders. Models like BLIP-2 (Li et al., 2023) and InstructBLIP (Dai et al., 2023) employ a Q-Former to bridge the gap between vision encoders and frozen LLMs. MiniGPT-4 (Zhu et al., 2023) introduces a combination of Q-Former and a linear projector to align vision and LLM modules more effectively.

In contrast, LLaVA (Liu et al., 2024b) utilizes a straightforward MLP projector to enhance the alignment between the vision encoder and LLM. mPLUG-Owl (Ye et al., 2023a) takes a different approach by fine-tuning the vision encoder first and then using LoRA (Hu et al., 2021b) to adjust the LLM. The Qwen-VL model (Bai et al., 2023) increases the visual module's resolution to 448 in order to improve visual processing performance. Alongside these open-source advances, proprietary models such as GPT-4V/4o (OpenAI, 2024; Achiam et al., 2023), Gemini (Team et al., 2023) and Qwen-VL-Plus/MAX (Bai et al., 2023) have demonstrated outstanding performance in both benchmarks and real-world applications. Given the simplicity and effectiveness of the LLaVA architecture, we adopt a similar framework for our model design in this work.

### 2.2. Tabular Understanding

Traditional tabular data learning primarily addresses standard tasks such as classification and regression (Ye et al., 2024; Meng et al., 2024; Ye et al., 2025b; Jiang et al., 2025). In recent years, increasing attention has been devoted to tabular data learning in open-world settings (Jiang et al., 2024; Cai & Ye, 2025; Hollmann et al., 2025; Liu & Ye, 2025; Ye et al., 2025a). Among these, tabular understanding involves comprehending the information contained within tabular data and can be broken down into several tasks, such as Table Structure Recognition (TSR) (Schreiber et al., 2017; Salaheldin Kasem et al., 2024), Table Detection (TD) (Gilani et al., 2017; Li et al., 2020), and Table Question Answering (TQA) (Chen et al., 2020; Talmor et al., 2021; Jin et al., 2022). Traditional methods, whether OCR-based (Appalaraju et al., 2021; Da et al., 2023; Gu et al., 2022) or OCR-free (Nassar et al., 2022; Kim et al., 2021; Feng et al., 2023; Wan et al., 2024; Zhao et al., 2024), have made significant strides in recognizing the structure and content of tables. However, we focus on more challenging TQA tasks.

Previous methods have tried a lot but the performance is not ideal. Donut (Kim et al., 2021) proposes a new task and a synthetic document image generator to pre-train the model to mitigate the dependencies on large-scale real document images Monkey and TextMonkey (Li et al., 2024; Liu et al., 2024c) adopt shifted window attention and use similarity to filter out significant tokens without redundancy. They expand the model's capabilities to encompass text spotting and grounding, and incorporate positional information into responses to enhance interpretability. mPLUG-DocOwl (Ye et al., 2023b) adjusts mPLUG-Owl for OCR-free document understanding. Tabpedia (Zhao et al., 2024) constructs a low-resolution vision encoder and a high-resolution vision encoder, with a concept synergy mechanism for visual table understanding. Deng et al. (2024) focuses on exploring various table representations and prompts LLMs directly. In addition to these advantages, they often suffer from the limitations of the ViT architecture. The patch-based division of the image into blocks causes the model to ignore the inherent relationships in the table, and it also tends to overlook certain conditions in the question.

## 3. Analyze the Ability of MLLMs in Tabular Understanding

We first introduce some notations for MLLMs in tabular understanding tasks. Then, we analyze the performance of current MLLMs on existing benchmarks and examine the reasons behind their performance limitations.

### 3.1. MLLMs for Tabular Understanding

A notable contribution in the field of MLLMs is LLaVA (Liu et al., 2024a), which presents a straightforward yet efficient approach to align the vision encoder with a pre-trained LLM. Specifically, given an input image $\mathbf{X}_v$, LLaVA uses the pre-trained CLIP vision encoder ViT-L/14 $g$ (Radford et al., 2021) to extract visual features $\mathbf{Z}_v = g(\mathbf{X}_v)$. The LLM, which is denoted as $f_\phi(\cdot)$ parameterized by $\phi$, is then employed to generate textual embeddings $\mathbf{H}_q$ from instruction $\mathbf{X}_q$. To ensure alignment between the vision encoder and the LLM, LLaVA learns a projector, represented by a multilayer perceptron (MLP) denoted as $\mathbf{W}$, which transforms visual features $\mathbf{Z}_v$ into language embedding tokens $\mathbf{H}_v$. This enables the seamless integration of multimodal information within the LLM framework.

$$\mathbf{H}_v = \mathbf{W} \cdot \mathbf{Z}_v, \text{ with } \mathbf{Z}_v = g(\mathbf{X}_v). \qquad (1)$$

Then, by inputting $\mathbf{H}_v$ and $\mathbf{H}_q$ into the language model $f_\phi(\cdot)$, the model generates the response $\mathbf{X}_a = f_\phi(\mathbf{H}_v, \mathbf{H}_q)$.

In the tabular understanding task, particularly the TQA task we focus on, $\mathbf{X}_v$ represents the input table image, and $\mathbf{X}_q$ represents the question. Our goal is for the final answer, $\mathbf{X}_a$, to be as consistent as possible with the correct answer $\mathbf{Y}_a$.

$$\min_{f_\phi} \sum_{i=1}^{N} \ell(f_\phi(\mathbf{H}_v, \mathbf{H}_q), \mathbf{Y}_a) \, . \tag{2}$$

$\ell$ is the loss function that measures the discrepancy between prediction and ground-truth.

In our paper, based on the rows and columns of tabular data, we categorize TQA into four key aspects: understanding individual elements (IE), interpreting rows or columns (RC), comprehending compositional conditions (CC), and performing basic calculations or reasoning (CR). A more detailed introduction is as follows:

- **IE:** This refers to the task of understanding and extracting specific cell values within a table, such as identifying the value at a particular row and column intersection. For example, "What is Student A's math score?"
- **RC:** This involves comprehending specific samples or attributes within a table. For instance, "Which course does Student A have the highest score in?" or "Which student has the best math score?"
- **CC:** This pertains to understanding table content that satisfies compositional conditions. Examples include, "What is the math score of the student with the highest total score?" or "Among the top three students in total score, how many have an 'A' in physical education?"
- **CR:** This refers to performing basic calculations or logical reasoning on specific cell values within a table. For example, "How much higher is the total score of the top student compared to the lowest-scoring student?"

### 3.2. Analysis on MLLMs in Tabular Understanding

**Models.** We selected several widely used state-of-the-art MLLMs, covering different architectures, training methods, and data. These MLLMs include two proprietary ones, GPT-4o (OpenAI, 2024), and Qwen-VL-Max (Bai et al., 2023), and seven public ones, LLaVA1.6-Vicuna7B/13B (Liu et al., 2024a), Monkey (Li et al., 2024), TextMonkey (Liu et al., 2024c), mPlug-Owl (Ye et al., 2023a), DocOwl (Ye et al., 2023b), InstructBLIP-Vicuna7B (Dai et al., 2023), and Donut (Kim et al., 2021).

**Data.** We evaluated the aforementioned models on the widely used tabular understanding benchmark: WikiTable-Questions (WTQ) (Pasupat & Liang, 2015). In particular, we divided the test set of the entire WTQ into four categories based on the nature of the tables, including understanding individual elements (IE), interpreting rows or columns (RC), comprehending compositional conditions (CC), and performing basic calculations or reasoning (CR). For conve-

*Table 1.* Accuracy on four types of table understanding tasks. The results show a significant gap between open-source and closed-source MLLMs. Additionally, the performance on the compositional condition QA is generally poor across the board.

|  | IE | RC | CC | CR |
|---|---|---|---|---|
| LLaVA-7B | 0.37 | 0.28 | 0.03 | 0.05 |
| LLaVA-13B | 0.47 | 0.32 | 0.07 | 0.10 |
| Monkey | 0.52 | 0.37 | 0.17 | 0.22 |
| TextMonkey | 0.52 | 0.40 | 0.25 | 0.18 |
| mPlug-Owl | 0.11 | 0.08 | 0.14 | 0.05 |
| DocOwl | 0.55 | 0.50 | 0.20 | 0.22 |
| BLIP | 0.05 | 0.07 | 0.07 | 0.25 |
| Donut | 0.27 | 0.03 | 0.02 | 0.07 |
| GPT-4o | 0.88 | 0.80 | 0.47 | 0.77 |
| Qwen-VL-Max | 0.85 | 0.82 | 0.48 | 0.70 |

nience in statistics, we randomly sampled 60 questions from each category and evaluated them on this smaller dataset.

**Evaluation Protocol.** Since this is an open-ended question-answering process, directly assessing the correctness of the model's answers is challenging. Additionally, relying solely on GPT-4 to evaluate the correctness of the QA pairs is not ideal, as GPT-4 is not always accurate. For instance, in Figure 1, GPT-4 overlooked the condition of "negative". However, when we input the question, ground truth, and the generated answer into LLMs, such as Qwen2.5-72B-Instruct (Yang et al., 2024), the evaluation becomes more accurate compared to using MLLMs alone. This strategy ensures a more precise assessment of MLLMs by transforming multimodal questions into a single modality for evaluation, thus reducing the likelihood of misjudging the model's capabilities and resulting in a more accurate evaluation.

**Results.** Table 1 shows the performance of different MLLMs on the tiny tabular understanding dataset. MLLMs perform poorly overall, with a significant performance gap compared to human experts, where all tasks are easy for humans. For instance, nearly all open-source models achieve less than $55\%$ accuracy on the simple IE task, while closed-source models perform better, with GPT-4o reaching $88\%$, compared to just $55\%$ for the best open-source model. On more challenging CC and CR tasks, the performance gap between open-source and closed-source models becomes even more pronounced. Even within the open-source models, there is a significant performance disparity. Document models, such as DocOwl and TextMonkey, outperform the more general large models like LLaVA on all tasks.

Furthermore, we found that even the strongest model, GPT-4o, performs poorly on CC tasks. The performance of open-source models is even worse, achieving less than half the accuracy of closed-source models. However, despite being compositional condition tasks, the answers to these ques-

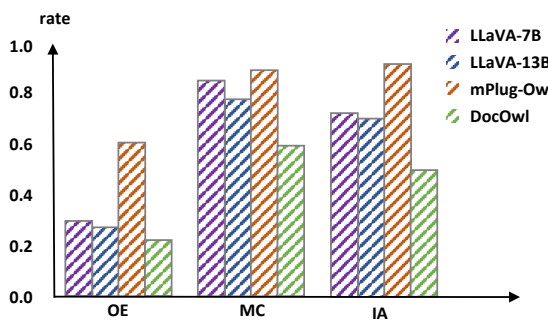

*Figure 2.* A further analysis of the compositional condition task. OE represents the OCR error rate, MC refers to the rate of missing conditions in answers, and IA denotes the incorrect alignment rate.

tions can be directly obtained from the table images, without requiring much additional knowledge. For human experts, these questions are also simple and straightforward. This raises the question: why do current large models struggle with even fundamental compositional condition tasks?

### 3.3. Why are MLLMs Limited in TQA?

To further investigate why MLLMs perform poorly on table understanding tasks, we analyze their structure, focusing on the visual encoder and LLM components. When it comes to tabular understanding, two key aspects arose: whether the table image can be accurately recognized and whether the LLMs can effectively comprehend the image. According to the results above, we primarily analyzed the performance of LLaVA-7B, LLaVA-7B, mPlug-Owl, mPlug-DocOwl.

**The adaptation of the visual encoder to table images is crucial.** We analyzed the capabilities of the vision encoder in MLLMs. We evaluated the instances of text recognition errors in the results of these MLLMs and found that LLaVA exhibited a significant number of text recognition mistakes. Even GPT-4o showed some cases of fuzzy text recognition errors. For example, they often misinterpret "1007" as "1001" or "1009". Additionally, while analyzing these results, we also observed errors in misalignment in rows or columns. For instance in Figure 1, the Oil Springs's change rate is -8.0, but recognized as Point Edward's change rate 0.1 by GPT-4o. When comparing the results of mPlug-Owl and DocOwl, we find that, with fine-tuning on document data, DocOwl significantly reduces rates of missing conditions and incorrect alignments. This highlights the importance of familiarizing the visual encoder with table images.

**Fully understanding each condition in the table is essential for successfully handling compositional condition tasks.** To further analyze why the model provides incorrect answers, we asked the model to answer step by step in a chain-of-thought format. We found that in conditional condition scenarios, the model may overlook certain conditions,

leading to incorrect final answers. As shown in the example above, even though the model initially considered the condition "most close to 0", it failed to factor the condition of "negative" when answering the question, resulting in an incorrect response. The results in Figure 2 show that open-source models like LLaVA and DocOwl frequently overlook conditions. Even GPT-4o overlooks certain conditions at a high rate. More analysis can be found in Appendix D.1

In summary, we find that the poor performance of MLLMs in tabular understanding can be attributed to two main aspects. First, the vision encoder has limited capabilities, failing to recognize the row and column information accurately. This is likely due to the visual encoder splitting the same sample or attribute into multiple patches, and relying solely on position embeddings may not capture this information effectively. Second, there is a gap in how MLLMs understand both the question and the image, which causes a disconnection between the two and leads to the neglect of some conditions mentioned in the question.

## 4. Compositional Condition Tabular Understanding

Motivated by our findings, this section presents our method, which incorporates row and column patches along with text-enhanced conditional tokens. We then outline the training process for our approach.

### 4.1. Contextual Information Extraction in Table Images

Traditional MLLMs, such as LLaVA, Donut, Tabpedia, typically use the ViT architecture as the vision encoder. Specifically, for 2D images, ViT reshape the image $\mathbf{X}_v \in \mathbb{R}^{H \times W \times C}$ into a sequence of flattened 2D patches $\mathbf{X}_v^p \in \mathbb{R}^{N \times (P^2 \cdot C)}$. where $(H, W)$ is the resolution of the original image, $C$ is the number of channels, $(P, P)$ is the resolution of each image patch, and $N = HW/P^2$ is the resulting number of patches, which also serves as the effective input sequence length for the Transformer.

However, when dealing with table images, using the same patch division can result in the splitting of certain information or column attributes across different patches, disrupting the contextual relationships within the table's rows and columns. For example, in the case in Figure 1, the village "Point Edward" and its corresponding change rate are divided into separate patches, severing their relationship within the same row. This causes incorrect matching during reading, where the change rate of "Oil Springs" is mistakenly paired with "Point Edward" instead. Therefore, ensuring that the model properly understands the content within the rows and columns becomes crucial.

To address this issue, we augment the original patches by

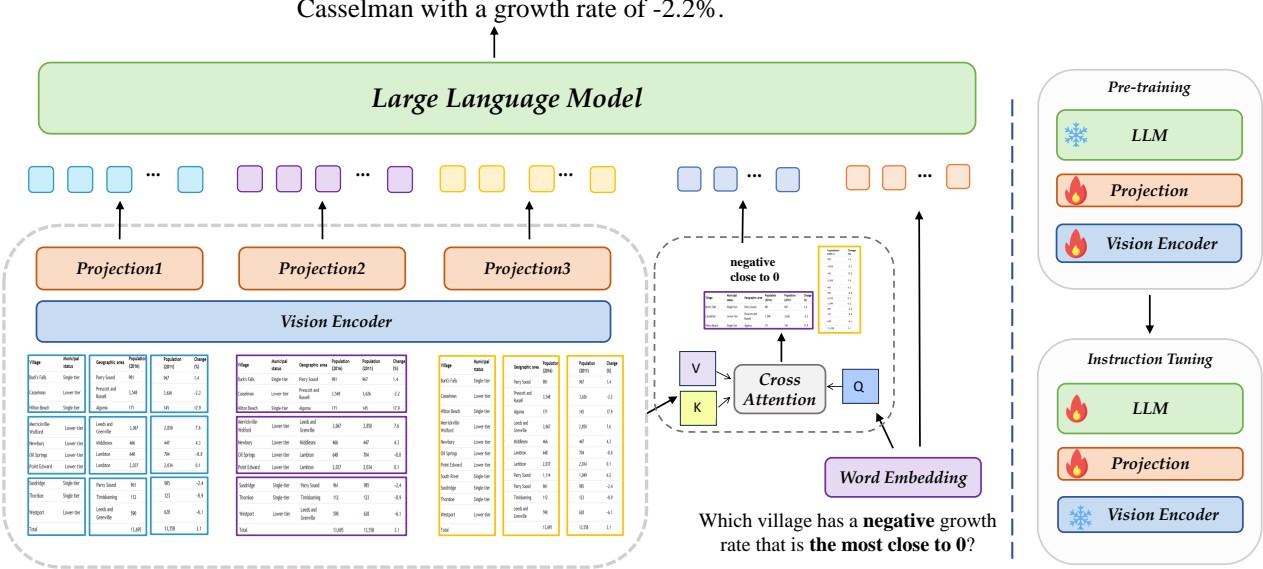

*Figure 3.* The overall architecture of CoCoTab. It primarily augments the table image with row and column patches and incorporates conditional tokens to enhance the relationship between each condition in the question and the image. The training details within each stage are presented on the right.

adding additional row and column patches. The goal is to better preserve the relationships within the sample or attributes through these row and column patches, thereby reducing the problem of misaligned rows in the vision encoder. Specifically, in addition to the original patches $\mathbf{X}_v^p$ with resolution $(P, P)$, we introduce row patches $\mathbf{X}_v^{row}$ with resolution $(P, W)$ and column patches $\mathbf{X}_v^{col}$ with resolution $(H, P)$. As a result, the total number of patches is $N = HW/P^2 + H/P + W/P$. Therefore, the final image tokens can be represented as follows:

$$\mathbf{Z}_v, \mathbf{Z}_{row}, \mathbf{Z}_{col} = g(\mathbf{X}_v^p, \mathbf{X}_v^{row}, \mathbf{X}_v^{col}). \tag{3}$$

After obtaining the tokens for all image patches, we learn a projection from each kind of patch token type to text, resulting in $\mathbf{H}_v = (\mathbf{H}_v^p, \mathbf{H}_v^{row}, \mathbf{H}_v^{col})$. By partitioning the table into row and column patches, we enhance the integration of sample-level and attribute-level information within individual patches, which preserves intra-sample and intra-attribute relationships, ensuring their structural integrity. Furthermore, the use of separate projections enables the learning of distinct mappings, effectively mitigating information loss.

### 4.2. Attend Tables with Questions

In current MLLMs, such as LLaVA, the visual encoder $g$ is built on the Vision Transformer (ViT). When processing image data, the model divides the image into patches and employs multiple layers of multi-head self-attention to capture the relationships between these patches. Similarly, in the language component, like the Transformer block in LLaMA, the architecture typically combines multi-head attention and MLP layers. For text, only self-attention is computed to capture relationships within the sequence. Although the image patch tokens are projected into the same space as text tokens, this method appears inadequate for truly effective cross-modal understanding, leading to the missing of certain conditions in tabular understanding.

Therefore, to fully leverage the information from both the image and the question, we attend to the image according to the question. First, we extract visual features using the visual encoder $g$ and then project them into language embedding tokens $\mathbf{H}_v \in \mathbb{R}^{N_v \times C}$, where $N_v$ is the vision token length and $C$ is the embedding dimension. Similarly, we obtain the question embedding $\mathbf{H}_q \in \mathbb{R}^{N_q \times C}$ from the text input via a word embedding table. Subsequently, to guide the transformation of visual features towards the question, we employ a cross-modal attention to obtain $\mathbf{H}_v'$.

$$\mathbf{H}_v' = \text{Attention}(\mathbf{Q}, \mathbf{K}, \mathbf{V}) = \text{Softmax}\left(\frac{\mathbf{H}_q \mathbf{H}_v^T}{\sqrt{C}}\right) \mathbf{H}_v, \tag{4}$$

where $\mathbf{Q}$ equals the matrix $\mathbf{H}_q$. $\mathbf{K}$ and $\mathbf{V}$ are equivalent to $\mathbf{H}_v$, $\mathbf{H}_v' \in \mathbb{R}^{N_q \times C}$. This process allows the visual features to be dynamically adjusted based on the input questions.

After obtaining the question-guided image tokens, we feed them into the LLM alongside the text tokens. Specifically, by inputting $\mathbf{H}_v$, $\mathbf{H}_v'$ and $\mathbf{H}_q$ into the language model $f_\phi(\cdot)$, the model generates the response $\mathbf{X}_a = f_\phi(\mathbf{H}_v, \mathbf{H}_v', \mathbf{H}_q)$. This approach explicitly combines the information from both the question and the image, rather than leaving the LLM to learn the relationships between the image and text tokens independently. Doing so more effectively captures the conditions specified in the question.

### 4.3. Training Stages

Our goal is to enhance different aspects of the tabular understanding capabilities of MLLMs. The whole training procedure is divided into two distinct stages.

**Stage 1: Patch Learning and Modality Alignment.** In this stage, we keep most of the parameters in the vision encoder and the LLM weights frozen, focusing on optimizing the projectors to align the visual features $\mathbf{H}_v$ with the pre-trained LLM word embedding. We unfrozen the row and column embedding to capture the representation of the samples and attributes. Row tokens and column tokens are aligned to LLM word embedding with their own projection to capture corresponding information. To enhance the diversity of images, we extract a portion of data from LAION and CC12M and construct the caption data, along with some tabular datasets, including Pub1M (Smock et al., 2022), WTQ (Pasupat & Liang, 2015), NAT-QA [1], PlotQA (Wang et al., 2024b), OCR-VQA [2] (Mishra et al., 2019).

**Stage 2: Instruction Tuning for Tabular Understanding.** In the first stage, we learned the representations of the patches and their mapping to the text embeddings. In stage 2, we freeze the vision encoder weights while continuing to train the projection and LLM. Our primary goal in this stage is to refine the representations of rows and columns, facilitating the LLMs' adaptation to these representations. Specifically, we focus on learning the row and column patch tokens and the image tokens that the questions attend.

To tackle the challenge of limited data in TQA, we adopt a semi-automatic approach to acquire image-QA data. We begin by randomly sampling tables from existing datasets, selecting a subset, and converting them into images. Next, we use GPT-4 to generate random questions about these tables and provide corresponding answers. After obtaining the initial QA data, we validate it using Qwen-VL-Max, retaining only the datasets that pass the validation. Recognizing that this step may introduce noise and potential errors, we apply a manual calibration process to further fine-tune and clean the data, thereby obtaining high-quality TQA data.

### 4.4. MMTU: A Massive Multimodal Tabular Understanding Benchmark

There are several existing tabular understanding benchmarks (e.g., WikiTableQuestions, TabFact, FinaQA, and ComTQA) for MLLMs, but they have some limitations: **(1) Narrow Domain.** FinaQA focuses primarily on simple calculations within the financial domain, TabFact assesses the truthfulness of content, and WTQ addresses basic ques-

---

[1] https://huggingface.co/datasets/staghado/ArXiv-tables

[2] https://huggingface.co/datasets/howard-hou/OCR-VQA

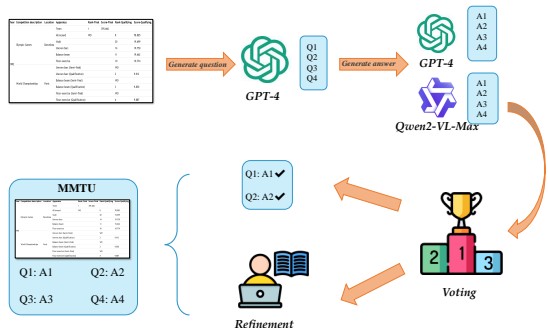

*Figure 4.* The calibration process for constructing the MMTU. The calibration process is mainly divided into three stages: GPT Generation, MLLMs Answer, and Manual Calibration.

tion answering. **(2) Uncertainty of Table Images.** Except ComTQA, other benchmarks do not provide table images. Since the method for converting data into table format can vary, this leads to potential biases in the evaluation results. **(3) Lack of Systematic Evaluation.** All existing benchmarks group similar QA tasks together without systematically evaluating specific capabilities, such as understanding individual cells, interpreting specific rows or columns, handling compositional conditions, and assessing reasoning and calculation abilities.

To address these challenges, we selected four tasks: understanding individual elements (IE), interpreting rows and columns (RC), comprehending compositional conditions (CC), and performing calculations or reasoning (CR). For the MMTU benchmark, we curated tables from WTQ (Pasupat & Liang, 2015), TabFact (Chen et al., 2019), and NAT-QA, creating four QA task types across over ten domains, yielding 8921 QA pairs. To ensure quality, GPT-4 generated questions, LLMs and human experts validated answers, retaining consistent pairs and resolving discrepancies, as shown in Figure 4. More details are in Appendix C.

## 5. Experiments

In this section, we first outline the experimental framework, providing details on the specific implementation, evaluation benchmarks, and MLLMs used for comparative assessment. Subsequently, we use tabular understanding benchmarks to conduct a comprehensive comparison of CoCoTab with state-of-the-art methods. Finally, this section summarizes the ablation study and visualizations for the tabular understanding case, highlighting CoCoTab's exceptional ability in handling compositional condition tasks.

### 5.1. Experimental Setup

**Implementation Details:** In this study, we configure Co-CoTab with the pre-trained Siglip-ViT (Zhai et al., 2023) as the vision encoder and Qwen2-Instruct (Yang et al., 2024)

*Table 2.* Accuracy performance comparison on existing benchmarks. MMTU is our proposed benchmark, with four aspects of tabular understanding, including understanding individual elements (IE), rows or columns (RC), compositional conditions(CC), and calculations or reasoning (CR). Our COCOTAB achieves the best performance.

| | MMTU | | | | WTQ | TabFact | ComTQA |
|---|---|---|---|---|---|---|---|
| | IE | RC | CC | CR | | | |
| LLaVA-1.6-7B (Liu et al., 2024a) | 0.50 | 0.32 | 0.12 | 0.06 | 0.23 | 0.27 | 0.27 |
| LLaVA-1.6-13B (Liu et al., 2024a) | 0.59 | 0.38 | 0.13 | 0.08 | 0.24 | 0.51 | 0.29 |
| Monkey (Li et al., 2024) | 0.39 | 0.24 | 0.28 | 0.06 | 0.23 | 0.51 | 0.19 |
| TextMonkey (Liu et al., 2024c) | 0.62 | 0.36 | 0.28 | 0.06 | 0.28 | 0.37 | 0.25 |
| mPlug-Owl (Ye et al., 2023a) | 0.11 | 0.08 | 0.15 | 0.06 | 0.10 | 0.50 | 0.07 |
| Docowl (Ye et al., 2023b) | 0.65 | 0.45 | 0.26 | 0.07 | 0.33 | 0.61 | 0.31 |
| shareGPT4V (Chen et al., 2025) | 0.17 | 0.09 | 0.16 | 0.04 | 0.13 | 0.52 | 0.10 |
| VisCPM (Hu et al., 2023) | 0.04 | 0.03 | 0.27 | 0.04 | 0.09 | 0.36 | 0.04 |
| InstructBLIP (Dai et al., 2023) | 0.06 | 0.04 | 0.08 | 0.04 | 0.09 | 0.51 | 0.04 |
| Donut (Kim et al., 2021) | 0.62 | 0.14 | 0.03 | 0.02 | 0.10 | 0.02 | 0.24 |
| COCOTAB | **0.68** | **0.50** | **0.43** | **0.38** | **0.45** | **0.74** | **0.34** |

as the backbone for LLM. The initial learning rates for the two stages are set as 2e-4 and 2e-6, respectively, with the batch size of 64 and 32. The learning rate for the vision encoder is set as 5e-7. The entire training process is about 5 days on the four A800 GPUs setup. Additionally, BF16 and TF32 precision formats are employed to balance speed and accuracy throughout the training process meticulously. As shown in Figure 3, we set three projections for visual patches, row patches, and column patches respectively.

**Evaluation Benchmark:** Our evaluation is divided into two parts: one assesses the capabilities of MLLMs in various aspects of table understanding, while the other evaluates their overall performance. The first evaluation is conducted on our proposed benchmark, MMTU, where we break down table understanding into four distinct components for assessment. For the overall performance, the evaluation covers a wide broad range of tabular understanding tasks, such as WTQ (Pasupat & Liang, 2015), TabFact (Chen et al., 2019), and ComTQA (Zhao et al., 2024).

**Comparison Models:** For comprehensive comparisons, we select leading open-source models in MLLMs, including LLaVA (Liu et al., 2024a), Monkey (Li et al., 2024), TextMonkey (Liu et al., 2024c), mPlug-Owl (Ye et al., 2023a), DocOwl (Ye et al., 2023b), ShareGPT-4V (Chen et al., 2025) InstructBLIP-Vicuna7B (Dai et al., 2023), VisCPM (Hu et al., 2023), and Donut (Kim et al., 2021).

### 5.2. Results

**Main Reuslts.** In this section, we present a table showcasing the table understanding capabilities of most current models. As shown in Table 2, nearly all open-source models perform modestly on table understanding tasks. Especially in compositional condition tasks, all open-source models

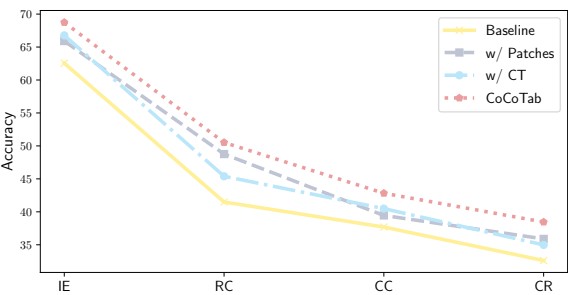

*Figure 5.* Ablation study of components in COCOTAB on MMTU.

perform poorly. Compared to LLaVA-7B, despite the use of a larger language model in LLaVA-13B, there is no significant improvement in performance on the CC and CR tasks. This validates our point that the primary limitation of the model's performance lies in the shortcomings of the visual encoder. In contrast, our COCOTAB achieves the best performance across all TQA tasks. Furthermore, it also delivers optimal performance on other existing benchmarks. In the CC and CR tasks, our performance far surpasses that of the original LLaVA, approaching or even close to the performance of the closed-source model Qwen2-VL in Appendix D.2. The experiment demonstrates the effectiveness of our approach to open-source MLLMs.

**Ablation Study.** To validate the importance of each component in our approach, we evaluated different models on the same dataset. "Baseline" refers to the original LLaVA model in our data, "w/ CT" indicates the addition of conditional tokens on top of the baseline, and "w/ Patches" represents the incorporation of row and column patches into the baseline. COCOTAB is our method, which combines both components. As shown in Figure 5, each component of our method contributes to an improvement over the baseline, and our approach achieves optimal performance. This not

| season | coach | overall | conference | standing | postseason |
|--------|-------|---------|------------|----------|------------|
| 2008--09 | anthony grant | 24-10 | 14-4 | 1st | ncaa first round |
| 2009--10 | shaka smart | 27-9 | 11-7 | 5th | cbi champion |
| 2010--11 | shaka smart | 28-12 | 12-6 | 4th | ncaa final four |
| 2011--12 | shaka smart | 29-7 | 15-3 | 2nd | ncaa third round |
| 2012--13 | shaka smart | 27-9 | 12-4 | 2nd | ncaa third round |

Q: What is the overall record for the 2011-12 season?
A: The overall record for the 2011-12 season is 29-7.

Q: Which season ended with an NCAA Final Four appearance?
A: 2010-11

Q: How many seasons had an overall record with more than 27 wins?
A: Two seasons had an overall record with more than 27 wins: \n\n- 2010–11 with a record of 28–12\n- 2011–12 with a record of 29–7

Q: What is the difference in conference wins between the 2008-09 and 2011-12 seasons?
A: The difference in conference wins between the 2008-09 season (14 wins) and the 2011-12 season (15 wins) is 1 win. This can be found in row 2008-09 and 2011-12 and column conference.

*Figure 6.* A case of COCOTAB. We present an example illustrating four types of questions on a single table image: understanding individual elements (IE), rows or columns (RC), compositional conditions (CC), and calculations or reasoning (CR). Our method demonstrates strong capability in effectively addressing all these question types.

only validates the effectiveness of row and column patches for table data but also confirms that adding attention between the question and the image enhances the model's ability to understand.

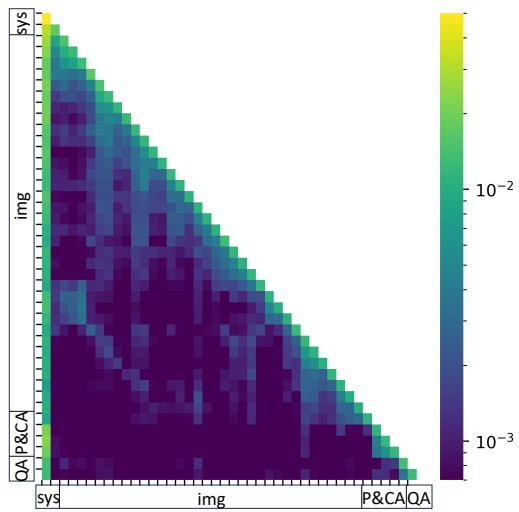

*Figure 7.* Visulization of COCOTAB. It can be observed that, after incorporating our row-column tokens and conditional tokens, the model places greater focus on relevant regions, leading to more accurate answer generation.

### 5.3. Visulization.

To validate the effectiveness of our approach, we visualize all tokens by generating attention matrices during the QA process. The results, depicted in Figure 7, reveal several key insights. While attention to image tokens becomes sparser, certain critical tokens continue to receive significant focus. Notably, the row-column tokens and conditional tokens we introduced emerge as pivotal in generating accurate responses, as they are consistently among the most heavily attended tokens. This visualization underscores the effectiveness of our row and column patches in extracting critical

information and aligning the model's focus with the specific conditions outlined in the questions. We also incorporate an example in Figure 6, showcasing our model's capability to address diverse types of TQA tasks effectively. More details can be found in Appendix D.5.

## 6. Conclusion

In this paper, we introduce COCOTAB, a new MLLM for TQA that effectively addresses compositional condition tasks. Our approach adapts to the features of table images by incorporating row and column patches in addition to the original patches. To help the model better understand each condition in the question, we use conditional tokens to enhance the interaction between the question and the image. Experimental results demonstrate that our method outperforms nearly all current open-source models. Through our experiments, we observe that current MLLMs still have limitations in handling compositional condition tasks and calculating or reasoning tasks. We hope this paper sparks further reflection on MLLMs, as even for foundational tasks, there is still substantial room for improvement.

## Acknowledgements

This work is partially supported by National Key R&D Program of China (2024YFE0202800), NSFC (62376118, 62476123), Key Program of Jiangsu Science Foundation (BK20243012), CCF-Tencent Rhino-Bird Open Research Fund (RAGR20240101), Collaborative Innovation Center of Novel Software Technology and Industrialization.

## Impact Statement

This paper presents work whose goal is to advance the field of Machine Learning, especially in Tabular Understanding. Our MMTU benchmark is designed to provide a standardized framework for converting structured tables of arbitrary formats into images. At the same time, it categorizes cur-

rent question-answering tasks into four types: understanding individual elements (IE), rows or columns (RC), compositional conditions (CC), and calculations or reasoning (CR). There are many potential societal consequences in our work related to our model and benchmark, none of which we feel must be specifically highlighted here.

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

# A. Training Datasets

In this section, we provide a comprehensive overview of the datasets employed in our study. During the first stage, we focus on two primary objectives: (1) effectively learning row and column patches, and (2) establishing a preliminary mapping from visual tokens to text embeddings. To achieve robust learning of row and column patches in tables, we trained our model using a diverse collection of table-based QA datasets. Additionally, we integrated the LLaVA-pretrain dataset to further enhance the training of the projection module. WTQ-Generate means we use the WTQ images and generate QA pairs with the help of GPT-4o. Moreover, the inclusion of conditional tokens during this stage allows the model to more effectively concentrate on visual tokens that are relevant to the posed questions, thereby improving its overall performance.

In the second stage, our primary objectives are to further refine the projection module and fine-tune the MLLM on table-based QA datasets. To achieve this, we leverage a range of TQA datasets, including WTQ, NAT-QA, and others. By fixing the visual tokens, we enable more effective optimization of the projection module, while unfreezing the LLM allows it to further adapt and learn from the QA datasets. Additionally, to enhance the model's versatility and performance across diverse tasks, we supplement the training with caption-based tasks, such as OCR-VQA and LLaVA-finetune. The datasets utilized in both stages are detailed in Table 3.

*Table 3.* Details on the CoCoTab's training data, derived from publicly available datasets and some TQA datasets.

| Training Stage | Datasets | Samples | Total |
|---|---|---|---|
| Stage 1 | LLaVA-1.5-pretrain (Liu et al., 2024b) | 50K | 1.4M |
| | Laion-Caption* (Schuhmann et al., 2022) | 50K | |
| | CC12M-Caption* (Changpinyo et al., 2021) | 50K | |
| | PUB-1M* (Smock et al., 2022) | 500K | |
| | WTQ* (Pasupat & Liang, 2015) | 14K | |
| | WTQ-Genrate* (Pasupat & Liang, 2015) | 142K | |
| | TabFact* (Chen et al., 2019) | 105K | |
| | NAT-QA* | 270K | |
| | Plot-QA* (Wang et al., 2024b) | 100K | |
| | OCR-VQA* (Mishra et al., 2019) | 168K | |
| Stage 2 | LLaVA-1.5-finetune (Liu et al., 2024b) | 100K | 1M |
| | PUB-1M* (Smock et al., 2022) | 82K | |
| | WTQ* (Pasupat & Liang, 2015) | 14K | |
| | WTQ-Genrate* (Pasupat & Liang, 2015) | 200K | |
| | TabFact* (Chen et al., 2019) | 105K | |
| | NAT-QA* | 174K | |
| | Plot-QA* (Wang et al., 2024b) | 100K | |
| | OCR-VQA* (Mishra et al., 2019) | 200K | |

# B. Training Details

As shown in Table 4, we provide the training hyperparameters for CoCoTab. Throughout all stages of training, we pre-train for one epoch and fine-tune for two epochs, with a batch size of 64 for the first stage and 32 for the second stage. We maintain an image resolution of 384x384 for all two stages and enable the gradient checkpoint mode for each training stage.

# C. MMTU: A Massive Multimodal Tabular Understanding Benchmark

In this section, we begin by addressing current benchmarks' shortcomings, and then by outlining the features that an optimal tabular understanding benchmark should have. Additionally, we propose and build a new benchmark along with its associated evaluation strategy.

*Table 4.* Training hyperparameters.

| Config | Stage 1 | Stage 2 |
|---|---|---|
| MLP expert network | 2 Linear layers with SiLU | |
| Deepspeed | Zero3 | Zero3 |
| Image resolution | 384×384 | |
| Image encoder | siglip-so400m-patch14-384 | |
| Feature select layer | -2 | |
| Image projector | 2 Linear layers with GeLU | |
| Epoch | 1 | 2 |
| Optimizer | AdamW | |
| Learning rate | 2e-4 | 2e-6 |
| Learning rate Vision | 5e-7 | - |
| Learning rate scheduler | Cosine | |
| Weight decay | 0.0 | |
| Text max length | 8096 | 2048 |
| Batch size per GPU | 16 | 8 |
| GPU | 4 × A800-80G | |
| Precision | Bf16 | |
| Gradient checkpoint | True | |

### C.1. Limitations of Existing Benchmarks

There are several existing tabular understanding benchmarks (e.g., WikiTableQuestions, TabFact, FinaQA, and ComTQA) for MLLMs, but they have some limitations: **(1) Narrow Domain.** FinaQA focuses primarily on simple calculations within the financial domain, TabFact assesses the truthfulness of content, and WTQ addresses basic question answering. **(2) Uncertainty of Table Images.** Except ComTQA, other benchmarks do not provide table images. Since the method for converting data into table format can vary, this leads to potential biases in the evaluation results. **(3) Lack of Systematic Evaluation.** All existing benchmarks group similar QA tasks together without systematically evaluating specific capabilities, such as understanding individual cells, interpreting specific rows or columns, handling compositional conditions, and assessing reasoning and calculation abilities.

### C.2. Characteristics of an Effective Tabular Understanding Benchmark

To more suitably evaluate the tabular understanding capabilities of MLLMs, an ideal benchmark should exhibit the following characteristics:

**(1) Multiple Domains of Content.** An excellent benchmark should cover a wide range of application domains to minimize bias that may arise in specific areas. By including data from multiple domains, we ensure that the model's performance is not influenced by any single domain. This not only enhances the benchmark's generalizability but also allows for a better evaluation of the model's cross-domain adaptability, ensuring that effective models are more widely applicable in real-world scenarios.

**(2) Unified Table-to-Image Conversion.** To ensure fairness in the benchmark, a standardized rule should be established for converting data in formats such as tables, CSV, HTML, and MD into images. This unified standard eliminates potential differences that could arise from using varying formats, thus preventing discrepancies in the conversion process from affecting the model's evaluation. By employing a consistent conversion method, all models being evaluated face the same challenges when processing data, thereby ensuring the evaluation results are more reliable and comparable.

**(3) Multidimensional Evaluation of Table Understanding.** Evaluating table understanding should not be limited to recognizing individual data cells but should encompass more complex tasks. For instance, the model needs to understand not only the content of individual cells but also the relationships between cells, rows, and columns. Additionally, the evaluation should assess the model's ability to understand compositional conditions, reasoning capabilities, and computational skills. This multidimensional assessment will comprehensively measure the strength of a table understanding model, ensuring that it remains efficient and accurate when handling complex data.

**(4) Accurate and Reasonable Evaluation.** When designing a benchmark, it is important to avoid including tables with excessive row-column differences that could lead to extreme cases. For example, tables with large row-column disparities

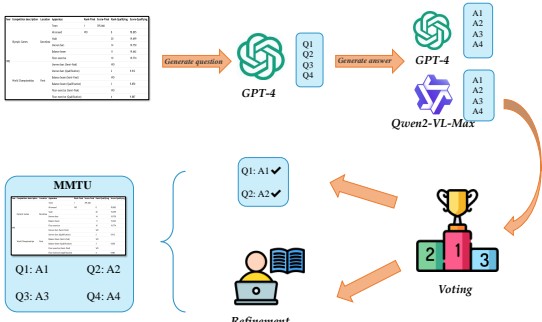

*Figure 8.* The calibration process for constructing the MMTU benchmark. The calibration process is mainly divided into two stages: GPT-4o Generation, MLLMs Answer, and Manual Calibration.

may have overly complex structures that hinder the model's performance, and should therefore be excluded from the benchmark's test scope. Additionally, the data and task design in the benchmark should be as clear and unambiguous as possible to ensure the reliability of the evaluation results. Most importantly, the dataset in the benchmark should not contain any errors, as any inaccurate data could lead to misleading evaluation results, ultimately distorting the true reflection of the model's performance.

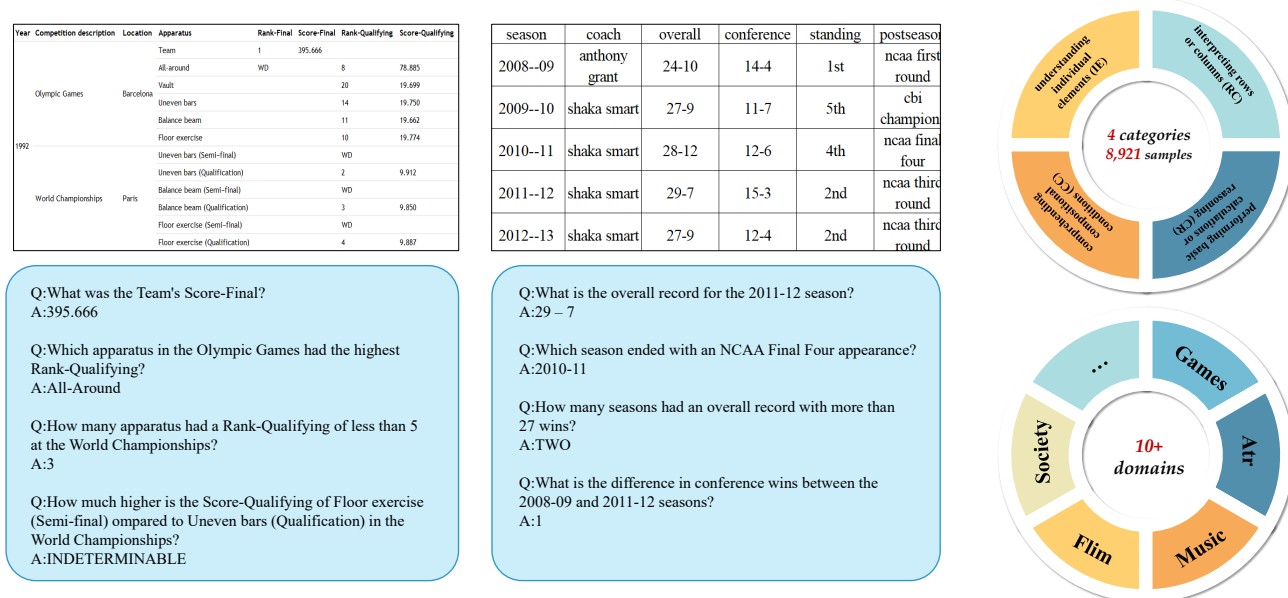

*Figure 9.* Detailed information and some cases in our MMTU benchmark.

## C.3. Construction of the MMTU benchmark

We select four tasks for inclusion: understanding individual elements (ie), interpreting rows or columns (rc), comprehending compositional conditions (cc), and performing basic calculations or reasoning (cr). For the massive multimodal tabular understanding benchmark, denoted as MMTU, we select and clean the suitable tables from WTQ, FinaQA and Arxiv papers. Based on these tables, we constructed four types of question-answering tasks according to different understanding objectives, resulting in four question categories, six domains, and approximately 10,000 question-answer pairs.

To reduce noise and errors that may arise during the data construction process, we implemented several strategies to improve translation quality, as shown in Figure 8. First, we used GPT-4 to generate questions for each image based on our predefined question types. Next, we enhanced the data quality through a peer evaluation process involving both large language models (LLMs) and human experts. Specifically, we leveraged three of the most reputable MLLMs in the industry: GPT-4, Qwen2-VL-Max, and InternVL2.5-78B to generate corresponding answers. Using consistency criteria, we retained

Table 5. Accuracy performance comparison on existing benchmarks.

| | MMTU | | | | WTQ | TabFact | ComTQA |
| | IE | RC | CC | CR | | | |
|---|---|---|---|---|---|---|---|
| LLaVA-1.6-7B | 0.50 | 0.32 | 0.12 | 0.06 | 0.23 | 0.27 | 0.27 |
| LLaVA-1.6-13B | 0.59 | 0.38 | 0.13 | 0.08 | 0.24 | 0.51 | 0.29 |
| Monkey | 0.39 | 0.24 | 0.28 | 0.06 | 0.23 | 0.51 | 0.19 |
| TextMonkey | 0.62 | 0.36 | 0.28 | 0.06 | 0.28 | 0.37 | 0.25 |
| mPlug-Owl | 0.11 | 0.08 | 0.15 | 0.06 | 0.10 | 0.50 | 0.07 |
| Docowl | 0.65 | 0.45 | 0.26 | 0.07 | 0.33 | 0.61 | 0.31 |
| shareGPT4V | 0.17 | 0.09 | 0.16 | 0.04 | 0.13 | 0.52 | 0.10 |
| VisCPM | 0.04 | 0.03 | 0.27 | 0.04 | 0.09 | 0.36 | 0.04 |
| InstructBLIP | 0.06 | 0.04 | 0.08 | 0.04 | 0.09 | 0.51 | 0.04 |
| Donut | 0.62 | 0.14 | 0.03 | 0.02 | 0.10 | 0.02 | 0.24 |
| CoCoTab | **0.68** | **0.50** | **0.43** | **0.38** | **0.45** | **0.74** | **0.34** |
| Qwen2-VL | 0.93 | 0.71 | 0.38 | 0.38 | 0.51 | 0.72 | 0.52 |
| GPT-4o | 0.95 | 0.88 | 0.61 | 0.85 | 0.67 | 0.73 | 0.62 |

QA pairs with matching answers. For pairs with inconsistent answers, we consulted multiple human experts to resolve discrepancies, resulting in the final MMTU benchmark.

This approach significantly enhanced the quality of the data, ensuring it is more representative of real-world scenarios and diverse across various question types. We list the detailed information and some cases in Figure 9. As a result, our benchmark provides a more comprehensive and accurate assessment of tabular understanding. We believe this improvement contributes positively to the overall robustness of our research findings.

# D. More Experiments

## D.1. More Analysis

In our analysis, we also identified significant challenges in the computational and reasoning capabilities of MLLMs. However, this paper does not focus on these issues for two main reasons. First, computational and reasoning limitations have long been a persistent challenge for large language models (Wei et al., 2022; Yuan et al., 2023; Yanid et al., 2024), primarily due to inherent architectural deficiencies. Second, our study prioritizes improving aspects of MLLMs that directly impact table comprehension, such as the limitations in visual encoding and the misalignment between visual and textual spaces. As a result, we do not directly address computational and reasoning challenges in this work.

## D.2. Comparison with GPT-4o

Furthermore, we incorporate closed-source methods in our benchmarks, including GPT-4o (OpenAI, 2024), and Qwen2-VL-Max (Wang et al., 2024a) to show the SOTA remarkable performance.

From the experimental results, a significant performance gap between open-source and closed-source models is evident. This discrepancy can be attributed to the following primary factors: Model Size: The parameter scale of models such as GPT and Qwen far exceeds that of our model, as their architectures are designed with substantially larger parameter counts. Training Data: Our model was trained on a relatively limited dataset, while these models likely benefited from extensive training on significantly larger and more diverse datasets, enabling them to achieve superior generalization and performance."

## D.3. Comparison with Structured Methods

Our method primarily focuses on question answering over visual tables, as in most cases, we do not have direct access to the structured information of the tables. Instead, we feed the table image directly into the model and train a more effective end-to-end model. This approach reduces the information loss between rows and columns that can occur during the intermediate step of converting the image into structured text, and also avoids potential errors introduced during such

Table 6. Comparison with structured methods.

| | IE | RC | CC | CR |
|---|---|---|---|---|
| TAPEX (Liu et al., 2021) | 0.73 | 0.79 | 0.33 | 0.19 |
| TaPas (Herzig et al., 2020) | 0.59 | 0.52 | 0.04 | 0.19 |
| COCOTAB | 0.68 | 0.50 | 0.43 | 0.38 |

Table 7. Inference time and memory cost of different methods.

| Model | Time(s) | Memory(G) |
|---|---|---|
| InstructBlip | 0.29 | 30.98 |
| LLaVA1.6 | 0.45 | 17.21 |
| DocOWL | 0.36 | 19.50 |
| TextMonkey | 1.68 | 23.96 |
| Donut | 0.42 | 0.478 |
| InternVL2.5 | 1.27 | 43.28 |
| TabPedia | 0.21 | 19.02 |
| CoCoTab | 0.20 | 20.47 |

conversions.

Here, we also conducted experiments using structured table parsing-based QA methods based on the data presented in Section 3.2. The experimental results are shown Table 6.

As seen from the results, our method achieves comparable performance to existing approaches on IE and RC tasks. However, for the more complex CC and CR tasks, COCOTAB significantly outperforms the structured table parsing-based QA methods. This is because once the table is converted into structured text, it becomes challenging for pure language models to preserve the spatial relationships between rows and columns, which leads to performance degradation on tasks that require complex reasoning, such as CC and CR. In contrast, our method directly processes the table image and, with the help of row and column patches, effectively captures the spatial layout and structure, enabling more accurate question answering. This further validates the effectiveness of our approach.

### D.4. Computation Cost

The original method divides the table image into 27×27 patches (729 total). COCOTAB adds 27 row patches (one per row, covering full width) and 27 column patches (one per column, covering full height), resulting in only 54 additional patches (7% increase over the original 729). This modest increase in token count (from 729 to 783) incurs minimal computational overhead during training and inference.

As shown in our ablation study, adding row/column patches alone improves accuracy on CC tasks over the baseline. This demonstrates that the cost increase is highly justified by the performance gains, particularly for critical tasks like compositional reasoning.

While our paper focused on accuracy and structural alignment improvements, we conducted additional experiments to quantify the computational overhead of COCOTAB. Key results and analyses are in Table 7.

We have measured the average inference time and memory consumption per sample. As shown in the results, our method does not introduce significant overhead. On the contrary, compared to LLaVA, our approach provides more concise and precise answers, leading to improved inference speed. COCOTAB adds only 7% in token count but outperforms in all baselines.

### D.5. More Visulizations

**Visualization.** To validate the effectiveness of our approach, we visualize all tokens by generating attention matrices during the QA process. The results, depicted in Figure 10, reveal several key insights. In the lower layers of the model, attention is

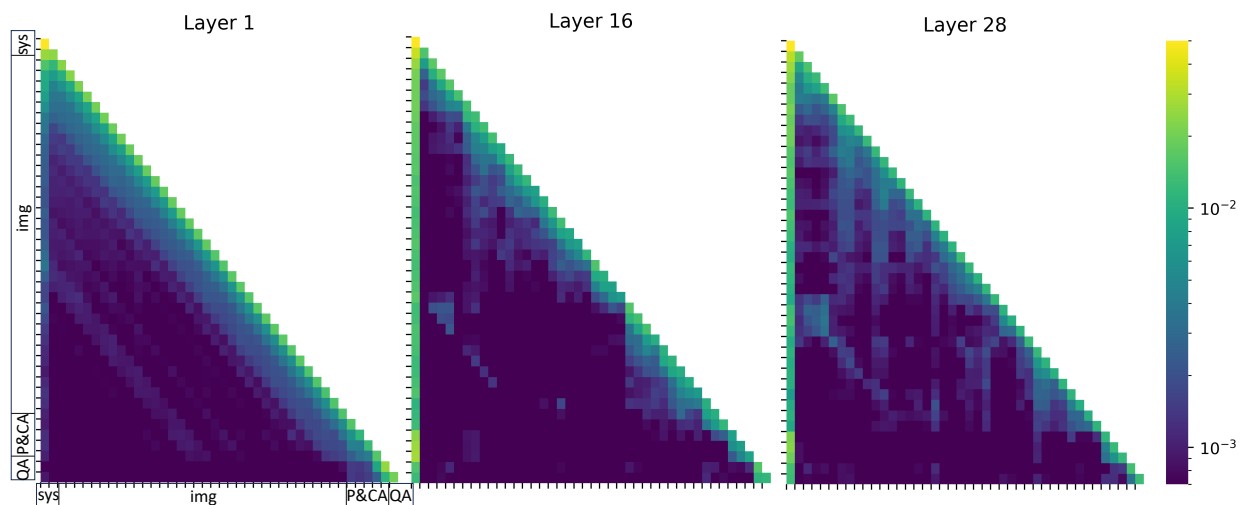

*Figure 10.* Visulization of attention matrix from lower layers to deeper layers.

distributed relatively evenly across different token types. However, in the deeper layers, while attention to image tokens becomes sparser, certain critical tokens continue to receive significant focus. Notably, the row-column tokens and conditional tokens we introduced emerge as pivotal in generating accurate responses, as they are consistently among the most heavily attended tokens. This visualization underscores the effectiveness of our row and column patches in extracting critical information and aligning the model's focus with the specific conditions outlined in the questions. We also incorporate several illustrative examples in Figure 11, showcasing our model's capability to address diverse types of TQA tasks effectively.

| Year | Competition description | Location | Apparatus | Rank-Final | Score-Final | Rank-Qualifying | Score-Qualifying |
|---|---|---|---|---|---|---|---|
| 1992 | | | Team | 1 | 395.666 | | |
| | | | All-around | WD | | 8 | 78.885 |
| | Olympic Games | Barcelona | Vault | | | 20 | 19.699 |
| | | | Uneven bars | | | 14 | 19.750 |
| | | | Balance beam | | | 11 | 19.662 |
| | | | Floor exercise | | | 10 | 19.774 |
| | | | Uneven bars (Semi-final) | | | WD | |
| | | | Uneven bars (Qualification) | | | 2 | 9.912 |
| | World Championships | Paris | Balance beam (Semi-final) | | | WD | |
| | | | Balance beam (Qualification) | | | 3 | 9.850 |
| | | | Floor exercise (Semi-final) | | | WD | |
| | | | Floor exercise (Qualification) | | | 4 | 9.887 |

Q: What was the Team's Score-Final?
A: The Team's Score-Final was 395.666.

Q: Which apparatus in the Olympic Games had the highest Rank-Qualifying?
A: All-Around. This can be found in Rank-Qualifying column.

Q: How many apparatus had a Rank-Qualifying of less than 5 at the World Championships?
A: At the World Championships in Paris, three apparatus had a Rank-Qualifying of less than 5:\n\n1. Uneven bars (Qualification) - Rank-Qualifying: 2\n2. Balance beam (Qualification) - Rank-Qualifying: 3\n3. Floor exercise (Qualification) - Rank-Qualifying: 4

Q: How much higher is the Score-Qualifying of Floor exercise (Semi-final) ompared to Uneven bars (Qualification) in the World Championships?
A: The Score-Qualifying of the Floor exercise (Semi-final) is not provided.

| season | coach | overall | conference | standing | postseason |
|---|---|---|---|---|---|
| 2008--09 | anthony grant | 24-10 | 14-4 | 1st | ncaa first round |
| 2009--10 | shaka smart | 27-9 | 11-7 | 5th | cbi champion |
| 2010--11 | shaka smart | 28-12 | 12-6 | 4th | ncaa final four |
| 2011--12 | shaka smart | 29-7 | 15-3 | 2nd | ncaa third round |
| 2012--13 | shaka smart | 27-9 | 12-4 | 2nd | ncaa third round |

Q: What is the overall record for the 2011-12 season?
A: The overall record for the 2011-12 season is 29-7.

Q: Which season ended with an NCAA Final Four appearance?
A: 2010-11

Q: How many seasons had an overall record with more than 27 wins?
A: Two seasons had an overall record with more than 27 wins: \n\n- 2010–11 with a record of 28–12\n- 2011–12 with a record of 29–7

Q: What is the difference in conference wins between the 2008-09 and 2011-12 seasons?
A: The difference in conference wins between the 2008-09 season (14 wins) and the 2011-12 season (15 wins) is 1 win. This can be found in row 2008-09 and 2011-12 and column conference.

*Figure 11.* Some cases of COCOTAB.

## E. Limitations and Future Works

In this paper, we aim to address the problem of tabular understanding. We propose four types of issues in table understanding, namely understanding individual elements (IE), interpreting rows and columns (RC), comprehending compositional

conditions (CC), and performing basic calculations or reasoning (CR). In our work, we primarily focus on addressing the CC problem, as it involves a deeper understanding of the intrinsic row-column structure of tables. While we propose four types of problems, we only partially tackle the CR problem. Through analysis, we find that a better understanding of the table's intrinsic structure can enhance the CR capabilities of MLLMs. However, the CR ability primarily places higher demands on the reasoning and computational power of LLMs, rather than on table understanding. Therefore, we were unable to fully address this issue in the current work, and it will be part of our future research.

From another perspective, there is still a significant gap in capabilities between open-source and closed-source models. As we analyzed earlier, this gap primarily stems from two factors: the difference in model size and data volume. Closed-source models tend to use larger parameter counts and more data, which ultimately contribute to their high performance. However, in our work, we have used relatively fewer parameters and less data, yet we have achieved better results with open-source models. In the future, we will further explore the gap between open-source and closed-source models in table understanding, investigating more fundamental issues, with the aim of narrowing this gap and even surpassing closed-source models.

Finally, we hope that this paper will spark deeper reflection on the fundamental capabilities of MLLMs, encouraging the exploration of basic yet important issues. We aim to contribute more meaningful work to the field of MLLMs.

