# OpenReview forum: "Compositional Condition Question Answering in Tabular Understanding"
_ICML.cc/2025/Conference — ICML 2025 poster_

### Official Review · Reviewer_qd5K · 2025-03-13

**Overall Recommendation:** 2

**Summary:**

This paper mainly aims at the problem of poor tabular comprehension ability of MLLM, and proposes a new method for extracting visual context information. By adding row and column-based image patch segmentation, and using the cross-attention method to fuse visual and textual features, the MLLM can capture the contextual relationships at the row and column levels in the table and the connections between images and text, thereby improving its performance on tabular Q&A tasks. At the same time, this paper also proposes a multimodal tabular understanding benchmark MMTU, including four tabular understanding tasks across ten different domains, to better evaluate the tabular understanding ability of MLLM.


## update after rebuttal
I still maintain my original decision because I think there are some fatal problems with this paper that have not been resolved.
1. The introduction of additional row and column patches does bring many extra visual tokens. This is contrary to the current development direction of MLLM. For high-resolution images, this method has obvious limitations. Moreover, there are already many redundant visual tokens in the current MLLM, and this method further increases the redundancy.
2. The authors configure COCOTAB with the pre-trained Siglip-ViT as the vision encoder and Qwen2-Instruct as the backbone for LLM. However, in the ablation study, they use the vanilla LLaVA-1.6 as the baseline to verify the validity of each component. This operation is puzzling. The authors did not address this issue in their rebuttal. Why not use the same baseline?
3. The experiment content is too small, only including a table and a figure. More adequate experimental design and analysis are needed.
4. Compared with some recent MLLMs, the proposed model does not have a significant performance advantage.
In summary, I think the current version of this paper is not well prepared and does not meet the standards of ICML.

**Claims And Evidence:**

Yes

**Essential References Not Discussed:**

The author needs to add more papers on multimodal large models published in the second half of 2024.

**Experimental Designs Or Analyses:**

1. The proposed method increases the number of input image patches and the input context length of the LLM in the image encoder stage, which will introduce additional computational overhead in model inference, but the proposed method is not compared with other methods in terms of model inference speed and computational overhead in the experimental part.
2. The methods compared in the experimental section are outdated.
3. The overall results of the method proposed by the authors are relatively low in the current community.

**Methods And Evaluation Criteria:**

This paper proposes a new benchmark for MLLM tabular understanding, which includes richer domains and problem types, which is meaningful for the evaluation of MLLM tabular understanding ability.

**Other Comments Or Suggestions:**

The author used the original LLaVA as a baseline and added two components on top to prove their effectiveness. However, the performance of the original LLaVA is low and is far behind some of the latest MLLMs. The improvement on such a weak baseline does not necessarily prove that these components can bring new inspiration or help to the community. The author should verify it on a stronger baseline.

**Other Strengths And Weaknesses:**

My main concerns are about the issues mentioned above regarding experiments.

**Questions For Authors:**

In addition to splitting the input image with the size of H*W into patches of P*P, the input image is also slice according to the dimensions of H*P and P*W. The image encoder used in the article is ViT, which can only accept the input patch size with a fixed size of P*P, and the article does not mention how the image patches with the size of H*P and P*W are input into the image encoder.

**Relation To Broader Scientific Literature:**

None

**Theoretical Claims:**

Yes, the proposed claims are correct.

---

> ### Author Rebuttal · Authors · 2025-04-01
>
> We appreciate your recognition of our setting and method and your valuable comments. We will answer the questions below, and we hope this clears up your concerns.
>
> **Q1**: additional computational overhead
>
> **A1**:  We appreciate the need to rigorously evaluate computational efficiency. While our paper focused on accuracy and structural alignment improvements, we conducted additional experiments to quantify the computational overhead of CoCoTab. Here are the key results and analyses:
> | Model         | Time (s) | Memory (G) |
> |--------------|---------|-----------|
> | **InstructBlip** | 0.294 | 30.98|
> | **LLaVA1.6**     | 0.456 | 17.21|
> | **DocOwl**       | 0.3695| 19.50|
> | **TextMonkey**   | 1.684| 23.96|
> | **Donut**        | 0.425 | 0.478|
> | **internvl**     | 1.277 | 43.28|
> | **TabPedia**     | 0.201 | 19.02|
> | **CoCoTab**      | 0.209 | 20.47|
>
> We have measured the **average inference time** and memory consumption per sample. As shown in the results, our method does not introduce significant overhead. On the contrary, compared to LLaVA, our approach provides more **concise and precise answers**, leading to improved inference speed. As noted in response A2 to Reviewer eFpb, COCOTAB adds only 7% in token count but outperforms in all baselines.
>
> **Q2**: More baselines and results.
>
> **A2**: Thank you for your critical feedback. We appreciate the opportunity to clarify and strengthen our experimental comparisons. Below, we address each concern systematically:
>
> Our work focuses on document and tabular understanding, a subfield of MLLM that demands specialized structural alignment capabilities. To ensure fair and state-of-the-art comparisons, we selected the strongest open-source baselines published in 2024, including LLava1.6(baseline), TextMonkey, DocOwl2.These models represent the most advanced techniques in document/table understanding as of mid-2024.
>
> For completeness, we also compared it with InternVL2.5-8B and TabPedia (NeurIPS 2024).
> | Model| IE  | RC  | CC  | CR  |
> |---------|----|----|----|----|
> | InternVL| 0.68 | 0.48 | 0.25 | 0.6  |
> | TabPedia| 0.71 | 0.47 | 0.34 | 0.27 |
> | CoCoTab| 0.68 | 0.5  | 0.43 | 0.38 |
>
> Although our method may not outperform InternVL and TabPedia in certain aspects, it significantly **surpasses both in CC** performance. Moreover, our training data (1.8M) is considerably smaller than that of TabPedia (~4M) and certainly much less than InternVL. Despite this, our approach still achieves competitive performance. We believe these results conclusively demonstrate CoCoTab's advancements in tabular understanding and we promise we will add these comparisons in our final version.
>
> **Q3**: The improvement on weak baseline LLaVA.
>
> **A3**:  Thank you for raising this critical point. We appreciate the need to validate our method’s contributions against stronger baselines and would like to clarify our experimental design:
>
>  Our ablation studies (Section 5.2) intentionally used the original LLaVA **architecture (Vision encoder + MLP + LLM)** to isolate the impact of our proposed components (row/column patches and conditional tokens) under strictly controlled conditions. By keeping the LLM, training data, and hyperparameters identical, we ensure that performance differences directly reflect the effectiveness of our structural enhancements. This approach is standard in ablation studies to **disentangle contributions from architectural changes** versus other factors. Our ablation study shows that the performance gain validates the importance of our proposed components.
>
> From another perspective, our ablation baseline (LLaVA + our training data) already **achieves higher accuracy** than vanilla LLaVA-1.6, demonstrating that our curated training data itself improves performance. This validates the quality of our data pipeline even before adding structural components.
>
> **Q4**: How the image patches with the size of HP and PW are input into the image encoder.
>
> **A4**: We appreciate the opportunity to clarify our methodology for handling multi-scale patches.
>
> In our CoCoTab, we generate row-wise (HP) and column-wise (PW) patch embeddings in parallel with the standard PP patches using lightweight convolutional operations. Here are the key steps:
>
> The input image is resized to 384×384 (as in SigLIP) and split into 27×27 patches of size 14×14 (P=14) with a 2D convolutions with kernal size PP, following standard ViT processing. Similarly, we apply **2D convolutions with kernel sizes H×P (for rows) and P×W (for columns)** to the resized image. These operations are computationally lightweight and performed before the ViT’s patch embedding layer. This design allows the ViT to implicitly reason about row/column structures without violating its architectural constraints. We will revise Section 4.1 to detail these steps.
>
> Thank you again for highlighting this ambiguity. We hope this clarification helps strengthen the details of our proposed CoCoTab.

---

### Official Review · Reviewer_eFpb · 2025-03-13

**Overall Recommendation:** 3

**Summary:**

The authors identified two issues that the current models lack in the context of multimodal table understanding: 1) the visual encoder’s patch-based processing which splits rows and columns can lead to misalignment, and 2) a failure to fully integrate the conditions specified in the query with the visual features.

To tackle these issues, the authors propose CoCoTab: 1) augment the vision encoder with row and column patches, and 2) add conditional tokens to explicitly link query conditions to the corresponding image regions.

Finally, the authors propose the Massive Multimodal Tabular Understanding (MMTU) benchmark to assess the full capabilities of MLLMs in tabular understanding, and show that CoCoTab performs best.

**Claims And Evidence:**

The authors posit that current MLLMs underperform on compositional condition questions due to patch misalignment and the overlooking of query conditions. This is backed by initial analysis by splitting TQA into 4 aspects: IE, RC, CC, and CR and show that CC is particularly bad for most models. The main evidence other than the effectiveness of the proposed method (will discuss more in the later sections). In all, the claim/hypothesis is clear and straightforward and easy to test.

**Essential References Not Discussed:**

See Relation To Broader Scientific Literature.

**Experimental Designs Or Analyses:**

See Methods And Evaluation Criteria.

**Methods And Evaluation Criteria:**

Strength:
- The proposed method is simple and shown to be effective.

Weakness:
- The proposed method can be seen as a form of data augmentation. Despite the simplicity and effectiveness, this type of data augmentation might have the following limitations: 1) assumes a particular structure of the data (well structured table) and might not further transfer to more generic type of vision input (e.g., an image of tables drawn on a blackboard), 2) by adding more patches and training on them it increases the cost.
- The baselines comparisons are slim. How does this method compare with other data augmentation methods or position-aware encoding methods.
- The evaluation and comparison doesn't seem totally fair. The proposed method is trained on a couple of datasets (sec 4.3), some are of the same distribution w.r.t. the final evaluation (e.g, WTQ), but the other models in comparison are not further trained.

**Other Comments Or Suggestions:**

N/A

**Other Strengths And Weaknesses:**

The paper is clear and easy to follow. My concern is mostly around the baseline comparisons and the evaluation protocol (i.e., other models are not being trained, thus the comparison is not exactly fair).

**Questions For Authors:**

See previous sections.

**Relation To Broader Scientific Literature:**

The authors positioned the work in the MLLM and tabular understanding literature. But I find the discussion a bit lacking around the data augmentation aspect of the proposed method. I would love to see more comparison on this front.

**Theoretical Claims:**

The authors do not propose theoretical claims in this paper.

---

> ### Author Rebuttal · Authors · 2025-04-01
>
> We appreciate your recognition of our setting and method and your valuable comments. We will answer the questions below, and we hope this clears up your concerns.
>
> **Q1**: A well-structured table might not further transfer to a more generic type of vision input.
>
> **A1**: Thank you for this insightful observation. We fully agree that extending tabular understanding to unstructured or free-form visual inputs (e.g., handwritten blackboard tables) is an important and challenging direction. Our work focuses on structured tables for two key reasons:
>
> **Foundation for Real-World Applications**: Existing benchmarks (e.g., WTQ, TabFact) and practical use cases predominantly involve structured tables. Even in these "well-structured" settings, current MLLMs struggle with basic compositional reasoning. Addressing these fundamental limitations is a necessary step before extending to more complex, unstructured scenarios.
>
> **Generalization via Training Diversity**: While our primary evaluation focuses on structured tables, CoCoTab’s training incorporates diverse visual QA datasets (e.g., OCR-VQA, PlotQA) that include semi-structured inputs such as book covers, posters, and diagrams. These datasets help the model develop robustness to irregular layouts, partial occlusions, and noisy text—capabilities that can facilitate generalization to blackboard-style tables.
>
> We acknowledge that fully adapting to free-form tables requires additional innovations, such as dynamic patch resizing to accommodate arbitrary layouts. We will incorporate a discussion of these limitations and potential future directions in the final version.
>
> **Q2**: More patches and the cost.
>
> **A2**: Thank you for raising this practical concern. We acknowledge that introducing additional patches incurs computational overhead. However, our design carefully balances cost and performance:
>
> The original method divides the table image into 27×27 patches (729 total). COCOTAB adds 27 row patches (one per row, covering full width) and 27 column patches (one per column, covering full height), resulting in only 54 additional patches (7% increase over the original 729). This **modest increase in token count** (from 729 to 783) incurs minimal computational overhead during training and inference.
>
> As shown in our ablation study, **adding row/column patches alone improves accuracy** on CC tasks over the baseline. This demonstrates that the cost increase is highly justified by the performance gains, particularly for critical tasks like compositional reasoning.
>
> In summary, COCOTAB achieves **performance gain with only a marginal cost increase**, making it a practical and efficient solution. **Please refer to A1 to Reviewer qd5K for more details.**
>
> **Q3**: The baselines are slim.
>
> **A3**: Thank you for your feedback. Our work focuses on improving tabular understanding in MLLMs, especially for compositional QA. To ensure fair comparisons, we chose state-of-the-art document-understanding MLLMs that integrate advanced techniques like data augmentation and positional encoding. For example, Donut: Uses synthetic data for pertaining. DocOwl: Leverages high-resolution document images for positional awareness. TextMonkey: Utilizes shifted window attention for spatial understanding.
>
> These baselines represent the strongest open-source MLLMs in this area and inherently employ the techniques you mentioned. Yet, as shown in Table 2, CoCoTab outperforms them on all tasks, highlighting that existing approaches still **struggle with tabular structure alignment and conditional reasoning**. **More experiments are in A2 to Reviewer qd5K.**
>
> **Q4**: The evaluation and comparison don't seem fair.
>
> **A4**: We appreciate the importance of fair comparisons and would like to clarify our experimental design:
>
> **Strict Data Separation**: While WTQ was included in our training data for Stage 1, we rigorously ensured that the test splits of WTQ, TabFact, and other benchmarks were **never exposed during training**. Many compared models are pre-trained or fine-tuned **on the same data** (e.g., DocOwl uses DocVQA, TabFact and WTQ). Our work follows a similar protocol but focuses on structural alignment.
>
> **MMTU**: The proposed MMTU was designed to evaluate TQA from different aspects. It includes tables from WTQ but **reformulates questions and introduces new compositional conditions**, ensuring no overlap with training data. Previous work [1] shows that existing benchmarks lack variation in their questions, resulting in a bad performance with perturbations. CoCoTab's superiority on MMTU and ablation study further validates its generalizability.
>
> In summary, our experiments adhere to fair evaluation standards, and the improvements are attributable to our method’s architectural innovations rather than data bias. We appreciate your attention and we will make it clear in the final version.
>
> [1] CharXiv: Charting Gaps in Realistic Chart Understanding in Multimodal LLMs. NeurIPS, 2024.

---

### Official Review · Reviewer_R1mg · 2025-03-14

**Overall Recommendation:** 3

**Summary:**

COCOTAB is a novel method for improving table question answering, particularly for queries with compositional conditions. The paper identifies two key challenges in existing models: the disruption of table structure due to patch-based vision encoders and the models' tendency to miss important query conditions. To overcome these, COCOTAB introduces dedicated row and column patches to better preserve structural relationships and uses conditional tokens to align query conditions with visual features. Additionally, the paper presents the MMTU benchmark, a comprehensive dataset for evaluating various aspects of table understanding. Experimental results show that COCOTAB significantly outperforms prior approaches, especially on complex compositional tasks.

**Claims And Evidence:**

Most claims are supported by convincing experimental evidence. For instance, the paper clearly demonstrates through quantitative comparisons and ablation studies that existing MLLMs struggle with compositional conditions, and that augmenting the vision encoder with dedicated row and column patches—along with introducing conditional tokens—significantly improves performance on benchmarks like MMTU.

However, one potential concern is the claim that the vision encoder’s patch-based approach is solely responsible for the shortcomings. While evidence supports its major role, other factors (such as the LLM's reasoning and alignment capabilities) might also contribute, but these aren’t fully disentangled in the experiments. Overall, though, the primary claims are well supported.

**Essential References Not Discussed:**

I didn't notice any missing key references.

**Experimental Designs Or Analyses:**

The experimental design is sound overall. The paper evaluates models on table question answering by dividing tasks into four aspects—individual elements (IE), rows/columns (RC), compositional conditions (CC), and calculations/reasoning (CR)—and uses both the WikiTableQuestions (WTQ) dataset and the new MMTU benchmark. The authors also conduct detailed ablation studies (e.g., Figure 5) and error analyses (e.g., Figure 2) to pinpoint issues like misalignment and missing conditions.Yes. I reviewed the supplementary material.

**Methods And Evaluation Criteria:**

The proposed methods and evaluation criteria are well-aligned with the challenges of table question answering. The authors introduce row and column patches to preserve the structural relationships within tables and conditional tokens to ensure that query conditions are accurately attended. This dual approach directly targets the shortcomings of patch-based vision encoders and misaligned query interpretations.

On the evaluation side, breaking down table understanding into four aspects—individual elements (IE), rows/columns (RC), compositional conditions (CC), and calculations/reasoning (CR)—along with the introduction of the MMTU benchmark, offers a comprehensive framework.

**Other Comments Or Suggestions:**

Overall, the paper is well-written and clearly presents its contributions.

**Other Strengths And Weaknesses:**

Strengths and weaknesses were discussed in other sections.

**Questions For Authors:**

- While the paper emphasizes the vision encoder’s patch-based approach as a major contributor to misalignment, have you conducted experiments to isolate the impact of the LLM’s reasoning and alignment capabilities separately?
- Did you have some qualitative examples showing the failure cases of COCOTAB?

**Relation To Broader Scientific Literature:**

The paper builds on prior work in multimodal and document understanding by addressing shortcomings identified in existing MLLMs. Earlier studies (e.g., Donut, mPLUG-Owl, LLaVA) showed that patch-based vision encoders and general alignment methods work well for basic table extraction and OCR tasks, but they struggle with the inherent structure of tables when faced with compositional condition questions. COCOTAB extends these ideas by introducing row and column patches to preserve the table's spatial relationships and by incorporating conditional tokens to more effectively align the query with the table content.

**Theoretical Claims:**

The paper does not present formal proofs for its theoretical claims. Instead, it provides mathematical formulations (e.g., Equations 1–4) that describe the model's architecture and the transformation of visual tokens into language embeddings. These expressions are consistent with standard practices in multimodal models and transformer architectures. Since the focus is on designing and empirically validating a new method rather than on deriving novel theoretical guarantees, no issues were found in the provided formulations.

---

> ### Author Rebuttal · Authors · 2025-04-01
>
> We appreciate your recognition of our setting and method and your valuable comments. We will answer the questions below, and we hope this clears up your concerns.
>
> **Q1**: While the paper emphasizes the vision encoder’s patch-based approach as a major contributor to misalignment, have you conducted experiments to isolate the impact of the LLM’s reasoning and alignment capabilities separately?
>
> **A1**:  Thank you for raising this critical question. We fully acknowledge the challenge of disentangling the LLM’s reasoning capabilities from alignment issues in multimodal systems. To address this, we designed ablation studies (Section 5.2 and Figure 5) to **isolate the impact of our proposed structural enhancements** (row/column patches and conditional tokens). These performance gains with our architecture were achieved without modifying the LLM, demonstrating that our structural enhancements directly address vision-text alignment limitations, independent of the LLM’s reasoning capabilities.
>
>  We agree that stronger LLMs can further improve performance. However, our method focuses on optimizing the visual encoder and cross-modal interaction, which are **orthogonal to the choice of LLM**. This ensures compatibility with any downstream LLM.
>
> In summary, our ablation studies confirm that the performance improvements stem from better structural alignment in the visual encoder and cross-modal attention, rather than LLM-specific reasoning enhancements. Thank you for your valuable question. We promise that we will make this clear in the final version.
>
> **Q2**: Did you have some qualitative examples showing the failure cases of COCOTAB?
>
> **A2**: Thank you for this question. While COCOTAB improves performance on CC tasks, we acknowledge that failure cases persist, particularly in scenarios involving **OCR errors or complex reasoning**. Here are two representative examples from our analysis:
>
> In our response in CC task,  the model correctly identified relevant rows but erred in misrecognized scores (e.g., “93” misread as “98” in one cell). As shown in Appendix E, the OCR ability of our model may be strengthened through other strategies, such as more datasets and more vision encoders or tokens.
>
> From another perspective, when it comes to CR tasks, the model erred in computing the difference between 63.49 - 38.77 (= 24.72) as 37.22. In this paper, we concentrate on CC tasks. For such calculation and reasoning tasks, other methods such as Group Relative Policy Optimization (GRPO) in DeepSeek will help improve performance.
>
> Thanks for your valuable suggestion. We will discuss these failure cases in Appendix and plan to address them via OCR error correction and enhance the reasoning ability in future work.

---

### Official Review · Reviewer_9AZj · 2025-04-10

**Overall Recommendation:** 3

**Summary:**

This paper investigates the table understanding task under compositional condition questions using multimodal large language models (MLLMs). The authors point out that current MLLMs face two major challenges: the inability of the vision encoder to accurately recognize table row contents and the tendency of the model to overlook conditions specified in the question. To address these issues, the paper proposes a new method, COCOTAB, which enhances row/column patch information in the vision encoder and introduces conditional tokens between visual patches and queries to improve the model’s ability to understand tabular data. Additionally, the paper constructs the MMTU benchmark dataset to comprehensively evaluate the capabilities of MLLMs in table understanding tasks.

**Claims And Evidence:**

Yes

**Essential References Not Discussed:**

No

**Experimental Designs Or Analyses:**

The authors present a comparative evaluation across multiple state-of-the-art open-source and proprietary MLLMs, conduct a detailed error analysis (OCR, alignment, and condition misses), and perform ablation studies isolating each component of their method.

**Methods And Evaluation Criteria:**

The proposed method is reasonable and well-aligned with the problem. The authors augment ViT-based vision encoders with structural patch enhancements (row and column tokens) and apply cross-modal attention using conditional tokens to better ground queries to relevant table regions. The evaluation framework is clear and systematic, using a new benchmark (MMTU) with four distinct task types (IE, RC, CC, CR) alongside existing benchmarks such as WTQ, TabFact, and ComTQA.

However, no measurements of model complexity or inference latency are reported. This omission makes it difficult to assess the practical deployment potential of the proposed system.

**Other Comments Or Suggestions:**

No

**Other Strengths And Weaknesses:**

Strengths
（1）The paper identifies common errors in MLLMs when handling tabular data, such as misalignment of the vision encoder with rows/columns (Figure 1: Westport’s negative growth rate being misrecognized as Oil Springs) and the model’s tendency to overlook conditions (e.g., GPT-4o exhibiting a high error rate on CC tasks). The motivation is reasonable. （2）The results indicate that the proposed method achieves a decent improvement.

Weaknesses
（1）The core improvements of the paper (row/column patches + conditional tokens) mainly focus on enhancing the vision encoder, raising concerns about the originality of the approach. （2）The paper primarily compares MLLMs but does not include comparisons with structured table parsing-based QA methods. （3）Table 2 only reports accuracy but does not provide inference time data. If COCOTAB requires significantly more inference time to achieve better performance, its practical application value may be limited.

**Questions For Authors:**

1.Efficiency Consideration: Your method adds both row/column patches and conditional tokens. Have you evaluated the runtime or memory overhead this introduces during inference? Please include such results if available.
2.Comparison with Structured Methods: Have you considered comparing your method against strong structured table QA models (e.g., TAPEX, TaPas)? Even if they operate on different modalities, such a comparison would contextualize your model’s relative advantages and limitations.

**Relation To Broader Scientific Literature:**

The paper builds effectively upon recent advances in MLLMs for table and document understanding, referencing core works like LLaVA, BLIP, Donut, DocOwl, and more. It identifies a gap in MLLM performance on CC-style questions and introduces a novel approach to fill this gap.

**Theoretical Claims:**

There are no formal theoretical proofs in the paper.

---

### Decision · Program_Chairs · 2025-05-01

**Decision:**

Accept (poster)

**Comment:**

The paper proposes CoCoTab, a novel method addressing limitations of Multimodal Large Language Models (MLLMs) in handling compositional condition questions in tabular understanding tasks. Reviewers highlighted clear strengths, including well-motivated enhancements to the vision encoder via additional row and column patches and the introduction of conditional tokens for improved condition-query alignment. The proposed Massive Multimodal Tabular Understanding (MMTU) benchmark further enriches the evaluation landscape by rigorously assessing compositional reasoning abilities across diverse tabular tasks. Despite initial concerns raised regarding model efficiency, inference overhead, and comparisons to structured parsing methods, the authors' rebuttal provided thorough clarifications and supplemental experimental data demonstrating reasonable computational overhead and superior performance on compositional reasoning tasks relative to state-of-the-art baselines. Given these clarifications and empirical results, the paper constitutes a meaningful advancement in multimodal tabular QA. Based on the final rating from reviewers, I recommend a weak acceptance (low priority: accept if there is room in the program) for this submission.